# A Pilot Study of Integrated Digital Tools at a School-Based Health Center Using the RE-AIM Framework

**DOI:** 10.3390/healthcare13151839

**Published:** 2025-07-29

**Authors:** Steven Vu, Alex Zepeda, Tai Metzger, Kathleen P. Tebb

**Affiliations:** 1Joe C. Wen School of Population and Public Health, University of California, Irvine, Irvine, CA 92697, USA; 2The Los Angeles Trust for Children’s Health, Los Angeles, CA 90017, USA; a.zepeda@thelatrust.org; 3William Beaumont School of Medicine, Oakland University, Rochester, MI 48309, USA; tmetzger@oakland.edu; 4Department of Pediatrics, Division of Adolescent and Young Adult Medicine, University of California San Francisco, San Francisco, CA 94143, USA; kathleen.tebb@ucsf.edu

**Keywords:** adolescents and young adults, sexual and reproductive health, underserved communities, mobile health, school-based health centers, Health-E You app, RAAPS, RE-AIM framework

## Abstract

**Introduction**: Adolescents and young adults (AYAs), especially those from underserved communities, often face barriers to sexual and reproductive health (SRH). This pilot study evaluated the implementation of mobile health technologies to promote SRH care, including the integration of the Rapid Adolescent Prevention Screening^TM^ (RAAPS) and the Health-E You/Salud iTu^TM^ (Health-E You) app at a School-Based Health Center (SBHC) in Los Angeles using the RE-AIM (Reach, Effectiveness, Adoption, Implementation, Maintenance) framework. **Methods**: This multi-method pilot study included the implementation of an integrated tool with two components, the RAAPS electronic health screening tool and the Health-E You app, which delivers tailored SRH education and contraceptive decision support to patients (who were sex-assigned as female at birth) and provides an electronic summary to clinicians to better prepare them for the visit with their patient. Quantitative data on tool usage were collected directly from the back-end data storage for the apps, and qualitative data were obtained through semi-structured interviews and in-clinic observations. Thematic analysis was conducted to identify implementation barriers and facilitators. **Results**: Between April 2024 and June 2024, 60 unique patients (14–19 years of age) had a healthcare visit. Of these, 35.00% used the integrated RAAPS/Health-E You app, and 88.33% completed the Health-E You app only. All five clinic staff were interviewed and expressed that they valued the tools for their educational impact, noting that they enhanced SRH discussions and helped uncover sensitive information that students might not disclose face-to-face. However, the tools affected clinic workflows and caused rooming delays due to the time-intensive setup process and lack of integration with the clinic’s primary electronic medical record system. In addition, they also reported that the time to complete the screener and app within the context of a 30-min appointment limited the time available for direct patient care. Additionally, staff reported that some students struggled with the two-step process and did not complete all components of the tool. Despite these challenges, clinic staff strongly supported renewing the RAAPS license and continued use of the Health-E You app, emphasizing the platform’s potential for improving SRH care and its educational value. **Conclusions**: The integrated RAAPS and Health-E You app platform demonstrated educational value and improved SRH care but faced operational and technical barriers in implementing the tool. These findings emphasize the potential of such tools to address SRH disparities among vulnerable AYAs while providing a framework for future implementations in SBHCs.

## 1. Introduction

Many adolescents and young adults (AYAs) engage in sexual behaviors that increase their risk of sexually transmitted infections (STIs) and unintended pregnancy, often due to limited sexual health knowledge, stigma, and access to care [1]. Blacks and Hispanic women and girls have often come from underserved communities and experience disproportionately high rates of adverse sexual and reproductive health (SRH) outcomes [2]. Specifically, STI/HIV screening rates among AYAs vary, and significant disparities in SRH outcomes among these groups persist because of barriers such as stigma, limited information, distrust of the medical field, and discomfort in discussing SRH [3]. There are also persistent disparities in unintended pregnancy rates among AYAs. For instance, 42.9% of Hispanic births and 53.5% of Black births were unintended compared to 30.7% for non-Hispanic Whites [2]. Annual health screenings for a wide range of healthy and health-risk behaviors for AYAs is recommended to promote health and mitigate health risk; this includes screenings on health/nutrition, physical exercise, mental/behavioral health, substance use, and sexual and reproductive health (SRH), which includes a sexual history and care for the prevention and treatment of STIs, contraceptive care, and other family planning services [4,5]. 

School-Based Health Centers (SBHCs) offer a promising model for promoting AYA SRH because they are usually located on or near school campuses and can often provide confidential SRH services, which can increase access to care and help reduce health disparities [6]. Integrating digital tools into SBHCs can further improve point-of-care delivery by streamlining the screening process and providing an electronic platform, which AYAs find to be a less judgmental approach to addressing the psychological aspects of behavior compared to advice from a clinician [7,8,9]. For instance, electronic health risk assessments have been shown to increase AYAs’ disclosure of a range of sensitive health risk behaviors (i.e., sexual activity, substance use, etc.) [7,10]. Within the context of SBHCs, electronic risk assessments like the Rapid Adolescent Prevention Screening (RAAPS) have proven effective in encouraging AYAs to disclose depression and related risk behaviors [11,12]. RAAPS is an evidence-based electronic screening tool that includes 21 items to assess SRH risks, diet and exercise, substance use, depression and suicide, violence and safety, and unintentional injury. Upon completion, it generates a comprehensive summary to help clinicians guide discussions and care during the visit.

While health risk screenings are an important step in identifying health risk behaviors, clinicians often lack time and comfort to address the full range of recommended AYA health issues [3,13]. As a result, there has been an increase in mobile health (mHealth) interventions to support the health of AYAs; however, systematic reviews have found that few have examined efficacy, especially around behavior change, and there is limited use of mHealth interventions focused on behavior change in clinical practice [14,15,16,17]. Health-E You/Salud iTu^TM^ (Health-E You) is one such mHealth application (app) that, in a cluster randomized control trial, significantly improved patient–clinician communication and increased contraceptive use by 118% [18]. The Office of Population Affairs has designated the Health-E You app as an evidence-based pregnancy-prevention tool. There is a significant gap in the literature examining the implementation of evidence-based interventions to improve the delivery of clinical care and patient outcomes.

This study sought to pilot test the implementation of an integrated tool that would first screen AYAs (using RAAPS: https://possibilitiesforchange.org) and then provide patients with a link to the Health-E You app (Drupal 7). We aimed to assess implementation facilitators and barriers using the RE-AIM (Reach, Effectiveness, Adoption, Implementation, Maintenance) framework [19,20,21]. See Figure 1 [19]. This pilot study builds on the prior successes of RAAPS and the Health-E You app, focusing on the implementation of both platforms at one SBHC in Los Angeles County. Investigating the barriers and facilitators to implementation will guide future efforts to expand the combined platform to other SBHCs and provide a template for integrating digital health tools in similar SBHC settings, ultimately improving health and educational outcomes for AYAs.

## 2. Materials and Methods

### 2.1. Setting/Context

The integration of RAAPS and the Health-E You app system was pilot-tested between April 2024 and June 2024 at one SBHC in Los Angeles County that expressed enthusiastic interest in using this integrated tool. This SBHC was opened in February 2023 to address the high unmet health needs of high school students in the eastern part of Los Angeles County. The SBHC serves adolescents who attend the school and most are from Latine/Hispanic backgrounds (92.6%), followed by Caucasian/White (6.1%), 0.4% African American/Black (0.4%), and Asian (0.3%) backgrounds. This SBHC provides comprehensive health care that includes primary care, general medicine, mental and behavioral health, and reproductive health and family planning services. This SBHC provides care within California, where it is legal for minors to consent to and receive sexual health services, including testing and treatment for sexually transmitted infections STIs, contraception, and other family planning services confidentially and without parental consent. 

### 2.2. Study Design

This study utilized a mixed-methods approach that included de-identified quantitative data gathered from RAPPS and the Health-E You app. Both apps included basic demographic information. However, Health-E You also gathered patients’ responses to a 7-item knowledge assessment, pre-app contraceptive use, app contraceptive recommendations, and the method the patient was most interested in using after completing the app. We also used a qualitative, case study design to evaluate the implementation of this novel tool using qualitative data gathered from post-implementation clinician interviews, two mid-implementation clinic observations, and notes from two technical assistance calls. The RE-AIM framework was used to evaluate the implementation process, as well as the potential for long-term sustainability, and consisted of the following components:

Reach: The total number of students who used the RAAPS screener and Health-E You app.

Effectiveness: The impact of the health risk screener and mHealth interventions on SRH service delivery.

Adoption: The extent to which RAAPS and Health-E You were incorporated into clinic practices.

Implementation: Major obstacles that hindered consistent usage.

Maintenance: Future outlook and sustainability of the system using feedback from clinic staff and clinicians.

### 2.3. Participants

All five clinic staff and clinicians involved in SRH care and behavioral health delivery were invited to participate in a 30 min semi-structured interview at the end of the implementation period to assess their attitudes about implementing the RAAPS and Health-E You app. To assess the reach of the app using the Re-Aim Framework, clinic staff were instructed to offer the evidence-based tool to all adolescent patients between 14 and 19 years of age. However, the inclusion criteria for analyses focused on sexually active females (sex assigned at birth) to assess pre-post contraceptive use. De-identified quantitative data from adolescent participants was extracted from app users and analyzed in aggregate post-implementation. Because this is an evidence-based practice tool that is being implemented in a real-world, SBHC setting, app users are not participating in a research study and their use of the app is part of regular clinic practice, so they did not require informed consent. Patients can decide whether or not they want to use these tools as part of their care. Verbal consent was provided by clinicians and staff to participate in the interviews. This study was approved by the University of California’s Institutional Review Board.

### 2.4. Intervention

Clinic staff were instructed to offer all adolescent patients the opportunity to use the electronic platform with its two components. The first part was RAAPS ^TM^, a cloud-based system used for adolescent risk screening. RAAPS is an evidence-based, comprehensive, standardized, and validated electronic health tool that assesses adolescent health risks for sexual and reproductive health, diet and exercise, substance use, depression and suicide, violence and safety, and unintentional injury [11,12]. Clinicians can log into the RAAPS system to view their patients’ responses. Once the patient completed the RAAPS screener, they were instructed to click on a link that routed them to the Health-E You app.

The Health-E You app has been described in detail elsewhere [8,18]. In brief, the app exists on the Drupal platform, which allows investigators to make some updates to the app without complicated programming. Patients first selected their language preference to use the app (English or Spanish) and answered a few basic demographic questions that routed them to tailored information. Specifically, non-sexually active females and males (both sex assigned at birth) were routed to a menu of various sexual health topics (e.g., what to expect in a sexual/reproductive health visit, condom use, HIV, Prep/Pep, pregnancy prevention, and/or emergency contraception). Sexually active females were routed to the pregnancy prevention/contraception decision support pathway. Health-E You customizes contraceptive recommendations by considering adolescents’ preferences, contraceptive history, satisfaction, adherence, etc. Users respond to a Likert agreement scale (4 = strongly agree, 3 = agree a little, 2 = disagree a little, and 1 = strongly disagree). Based on the user’s input, the app generates (a) recommendation(s) for non-barrier contraception and indicates the recommendation by placing a star next to the recommended method that is presented along the continuum of methods based on the World Health Organization’s (WHO) medical eligibility criteria for contraceptive use. At the same time, the app takes a reproductive justice approach and emphasizes that, regardless of the app’s recommendations, “the choice is yours” and encourages users to explore all of their options. The app also asks about potential contraindications and flags those for the clinician to discuss with the patient, highlights the importance of dual protection with condoms to reduce the risk of STIs and HIV, and educates users about emergency contraception. At the end of the app, a confidential summary of the patient’s Health-E You responses was sent automatically via secure e-mailed in real time for the clinician for review prior to the face-to-face encounter with the patient. The summary included basic health history information, what method of contraception they were currently using, what method(s) the app recommended, and what method they were most interested in using after they completed the app (which included the option, “I’m not ready to select a method”). Patients could access the Health-E You app without using the RAAPS screener and could use the RAAPS screener without using the Health-E You app. Both RAAPS and Health-E You can be used on any device (laptop, tablet, or smartphone).

### 2.5. Data Collection

Quantitative Data: De-identified data from the RAAPS and Health-E You app were provided to the research team for analysis.

Qualitative Data: Semi-structured interview guides were developed by the study team for the purpose of this implementation evaluation. Once trained, the research assistant conducted three interviews and the PI conducted two interviews in August 2024, after the completion of the study implementation, with clinic staff and clinicians involved in delivering SRH and behavioral health care at the SBHCs. The interviews were designed to examine the barriers and facilitators encountered during the implementation of the integrated web-based apps, staff perceptions of the platform’s usability, as well as feedback on the quality of support services provided throughout the study period. The interviews averaged 30 min and were completely confidential and voluntary. Participants were emailed a complimentary USD 30 digital gift card within two weeks following the interview. With confirmed verbal consent, the interviews were recorded and transcribed for thematic analysis. The interview questions gathered data on the following topics:

How the screener and app impacted clinic workflows, including the time required for students to complete the screening and the effect on clinical operations.

Any challenges that came up during the implementation period and, consequently, if the dedicated support services provided were adequate.

Subjective observations on how easy to use the app was for students of varying ages and grade levels.

The efficacy of the software in delivering real-time alerts to clinicians to be used during the consultation based on the students’ responses and activity.

Whether the screener and app enhanced the delivery of SRH services for students.

Whether the staff would like to see the platform renewed for next year and their reasoning.

In addition to the interviews, a member of the study team conducted two in-person, clinic observations mid-implementation and reviewed notes from three technical assistance calls to provide additional insights into implementation facilitators and barriers.

### 2.6. Data Analysis

The inclusion criteria for the analysis of app users were restricted to females (sex assigned at birth) who reported having sexual intercourse and who were between the ages of 14 and 19 years of age. Descriptive analyses for all quantitative data were conducted with SPSS version 29 (IMB, Chicago, IL, USA).

Responses to open-ended questions were coded and analyzed by frequency of themes. A coding guide was developed by the research team based on an initial review of responses. Two researchers (the PI and research assistant trained on the coding methodology) independently coded 10% of the responses in each category with an inter-coder reliability > 80%. After inter-coder reliability was established, the research assistant read through all the interview transcripts and coded the data, which was then used to conduct thematic analysis. The PI conducted random quality checks on the coding of the data. Common themes related to the implementation process such as barriers, facilitators, and staff perceptions of the platform’s impact on clinic operations were identified and discussed with all members of the research team. Discrepancies were resolved, and coding guidance was further clarified.

## 3. Results

### 3.1. Reach

During the implementation period from April 2024 to June 2024, there were 60 unique visits at the SBHC for adolescents between 14 and 19 years old. Of the 60 unique patients, 21 used the integrated RAAPS/Health-E You tool, while a total of 53 patients used only the Health-E You app. This represents a reach of 35.00% and 88.33%, respectively. All RAAPS participants completed the Health-E You app; however, not all Health-E You users completed the RAAPS. The racial/ethnic distribution of users was similar to the racial/ethnic distribution of clinic users, with the overwhelming majority coming from Latine/Hispanic backgrounds (Appendix A). All participants completed the app in English.

### 3.2. Effectiveness

The app is designed to provide patient-centered contraceptive decision-making support for sexually active females (sex assigned at birth), and non-sexually active females and males (sex assigned at birth) are routed to a menu of sexual health topics. Of the 53 users, 59.46% were sexually active females (sex assigned at birth). Prior to using the app, 23.08% were using a non-barrier contraceptive method (most were using only male condoms, inconsistently, and/or the withdrawal method). The app recommended a non-barrier method for all sexually active teens. After using the app, while most sexually active adolescents reported that they were not ready to select a method (57.14%), a total of 42.86% reported wanting to use a non-barrier method (see Appendix A for distribution of methods). The percentage increase of non-barrier contraceptive use prior to using the app (23.08%) and interest in using a non-barrier method after app use (42.86%) represents an 85.70% increase. Sexual health knowledge among sexually active females varied greatly by the seven “Myth Buster” pre-app use knowledge items with most correctly answering the item that an intra-uterine device (IUD) is easy for a clinician to insert and remove (83.33% correct); the pull-out method does not prevent pregnancy (75.0% correct) and birth control pills do not begin working immediately upon taking them (75.0% correct). Just over half (58.33%) correctly answered the question that contraception methods, especially IUDs, do not cause weight gain. Half (50.0%) correctly answered the question that birth control pills do not reduce STI risk and that LARCs do not make it more difficult to become pregnant in the future. Fewer (41.67%) correctly answered the question that decreased menstrual bleeding does not cause later health problems. Only four of the five adolescents rated the app using the five-star rating system prior to exiting the app. The mean rating was 4.75 stars.

### 3.3. Adoption

Clinicians and staff first received training on the use of RAAPS and the Health-E You app in April 2023, shortly after the SBHC opened in February. The clinic purchased iPads to use the technology in their SBHC. Refresher training sessions were provided in June 2023 and September 2023, focusing on both technical aspects and clinical integration. However, due to clinic start-up delays and clinic staff turnover, the implementation of the app was delayed until April 2024. At this time, two additional on-site trainings were provided to ensure clinic staff understood how to implement the technology. Staff were trained to first set up a confidential patient account in the RAAPS screening system so that the screening information could be shared with the clinician. Official implementation began in April 2024, when the necessary staff and in-person support were secured and trained, and implementation extended through the end of the academic year (June 2024).

### 3.4. Implementation

A thematic analysis of the semi-structured interviews revealed several key themes related to the implementation of the RAAPS and Health-E You app: value of the tool to improve patient care, workflow interference, resource limitations, screening length, challenges in therapeutic settings, two-step process confusion, alert delays, technical support, and perceived educational value. Each theme is described below. Many of the themes that emerged during the interviews were also observed during the two site visits to the clinic and were documented and addressed during the technical assistance calls.

#### 3.4.1. Perceived Educational Value

The clinic leadership and staff expressed that they valued the educational aspect of the Health-E You app, noting that it effectively covered SRH topics not included in standard questionnaires. The RAAPS screener was also seen as beneficial as it allowed students to disclose risky behaviors they might normally feel too uncomfortable to discuss face-to-face. Consequently, all interviewees agreed that they would like to see the license renewed for the following year (2024–2025). This is reflected in the following quotes:


*“I think it’s beneficial, because, like I said, this clinic does see a lot of kids and teenagers. Just so that they have that awareness. You know, it’s very weird for them to talk about sex or STDs or any of that stuff. But I think, you know, the awareness is definitely much needed and should be used.”*
—Family Nurse Practitioner


*“The app would really support adolescents with sexual health information, especially around contraceptive use, I think it is also of value to our clinicians.”*
—Chief Executive Officer

#### 3.4.2. Impact on Workflow

Clinic staff reported mixed findings that the RAAPS platform setup interfered with workflows. Some felt that for a small clinic like theirs, it was difficult having to manually log patients in with unique user IDs to enter into the platform. They often resulted in delayed appointments and added set-up time. Some staff found the setup to be challenging:


*“…because you first have to go online and log on the patient, and then get a username for them, and then log on. It’s a lot of steps to take for every patient… It would be very easy if you could just do it like it was an app, and you could just do it in the app altogether.”*
—Front Office Staff


*“It slows us down, that’s why we haven’t been able to do it as much. It takes up way too much time.”*
—Family Nurse Practitioner

However, once they were able to set it up and offer it to the adolescent patients, the process went smoothly:


*“When we do give students the survey, things run smoothly on both their end and ours.”*
—Front Office Staff

#### 3.4.3. Resource Limitations and Staffing

Due to limited staffing at the small SBHC, there were few available personnel to assist students with setup with the RAAPS platform or to monitor their completion of the screening to ensure students also use the Health-E You app. This is reflected in the following quotes:


*“I don’t have time to—it’s hard. We’re a smaller clinic… Family Nurse Practitioner*



*Maybe for a clinic with a bigger support staff. But we just don’t have the support staff.”*
—Family Nurse Practitioner


*“If the MA needs to, you know, like log in and at this location it is very small, so it’s just me in the front. One MA, and one provider. So, if we tend to be busy, it might be a little difficult for the MA to go online and check if they were done or not.”*
—Front Office Staff


*“We have had some staffing and provider change at the school and need to do additional trainings.”*
—Chief Executive Officer


*“We have had an increase in patients with no extra support and are unable to complete our core duties and log on to the Heath-E You, register the patients and give it to them to complete.”*
—Registered nurse

#### 3.4.4. Screening Length

Clinic staff expressed concerns regarding the time required to complete the RAAPS screener with the Health-E You app, which collectively took approximately 12.25 min (Appendix A). With average appointments lasting only 30 min, they felt that the length of the screening process significantly reduced time available for patient care. This is reflected in the following quotes:


*“It takes away from their 30-min sessions. So, if I have to cut it short, if they’re in the middle of, you know, processing something it takes away either time from them, or it’s going longer. And it’s now going into the next person’s session.”*
—Behavioral Health Provider


*“I’ve noticed that it takes some time. They take about 15 min to complete depending on if they have a topic they want to check out.”*
—Front Office Staff

#### 3.4.5. Two-Step Process Caused Some Confusion

The two-step format in the RAAPS and Health-E You app workflow created confusion among staff and app users. Following the completion of the RAAPS screener, students often did not realize there was a second component to complete, as the screen would reset to the start page without a clear prompt or indicator. This is reflected in the following quotes:


*“It has a weird thing where it looks like it takes them back to the first page after the first part, and they get frustrated. I get frustrated. I’m like, I don’t have time for this. I don’t want to do it.”*
—Family Nurse Practitioner


*“Sometimes they have to be prompted [to use the Health-E You app] because [when they complete the screener] they think it’s the end [and they don’t realize they need to click the link to use Health-E You].”*
—Front Office Staff


*“The only confusing part, too, is, I know there’s two parts to it. So, the kids who think that once they finish the first portion of it, they’re done. But I know there’s also the second part.”*
—Behavioral Health Provider

#### 3.4.6. Alert Delays

They reported delays in receiving alerts from the Health-E You platform. The alerts from Health-E You were sent via email but not integrated into the clinic’s primary electronic health system, which made it difficult to monitor in real-time, especially given limited staff resources. Consequently, responses were usually only reviewed at the end of the day, which created a risk of missing critical information that could have informed immediate clinical decisions. This is reflected in the following quotes:


*“Maybe it doesn’t register, or we have to wait a certain amount of time for it to actually show that we did the required tools on the computer… by that time we already move on to the next patient and so I can’t check the responses until the end of the day.”*
—Family Nurse Practitioner


*“If anything flags. I’m not going to see it, because the visits are already done. But you know I look at them at the end of the day to go through anything that I’m like–Oh, no, I need to address this like as A.S.A.P. [as soon as possible].”*
—Family Nurse Practitioner

### 3.5. Maintenance

#### 3.5.1. Needed Technical Support

Technical support and assistance were necessary as there were staffing changes and glitches with the technology. There was even one instance where prompt assistance was provided to staff to help them with navigating through the screener and app. This has implications for continued use of the app beyond the study period. This is reflected in the following quotes:


*“Someone came out to help troubleshoot that so that we can show them when the students complete this, it doesn’t take them to the next screen. So, we were showing them what it looked like for us.”*
—Behavioral Health Provider


*“I’d say technical support was pretty helpful for us, they do try to help with troubleshooting.”*
—Front Office Staff

#### 3.5.2. Desire to Continue Use the App

Most staff expressed a desire to continue to use the app. The CEO purchased the additional RAAPs license to continue to use the app beyond the study period.


*“I would like for them to extend it [continue using the app], just because there are some questions that if you know the student’s answer, we would be able to refer them to our therapist. So, I would definitely say that it is beneficial.”*
—Front Office Staff


*“I do see the value in it… I know I’ve been pointing out the bad things, but there are no other questionnaires that kind of address all the things that Health-E You does, and I do appreciate that.”*
—Family Nurse Practitioner

### 3.6. Clinic Observations and Technical Assistance Notes

The two-clinic observations indicated that staff occasionally would forget to offer the app to the adolescent patients; however, when offered, most adolescents agreed to use it. Sometimes patients were called back to the exam room before they completed the tool, and there was some confusion as to whether or not they should return the iPad to the front desk or take it with them to the exam room. No other challenges were observed.

Notes from the technical assistance calls were consistent with findings from the interviews. Specifically, they emphasized the importance of training new staff in how to use the technology and the need for this to be conducted by the study team. Further, there were some glitches in the technology and questions staff had—especially regarding registering patients using RAAPS and helping patients navigate as they move through the screener and app. In addition, we were informed that while they experienced challenges using the RAAPS, they offered patients the opportunity to use the Health-E You app in the interim, which aided in the interpretation of the quantitative findings that more patients used Health-E You than the combined tool.

## 4. Discussion

The findings of this pilot study highlight both the potential and the challenges of implementing a merged RAAPS and Health-E You app platform within one relatively newly established SBHC. While the clinic leadership, clinicians, and staff valued the educational content and ease of use for AYAs, they also reported several challenges in implementing this technology in real-world clinical practice. Specifically, staff mentioned the time-intensive screening process and technical issues such as delayed alerts. This is consistent with prior research on mHealth interventions, which frequently report logistical hurdles and technological integration issues [22]. There were also delays in the implementation of the app. This was in part due to typical challenges with starting up a new clinic; however, staff turnover played a critical role as well. To put this in context, clinics, especially safety-net clinics like SBHCs, often face challenges including burnout, stress, and the strain of managing increased workloads, and these challenges were exacerbated by the COVID-19 pandemic and continue to be a problem in the post-COVID-19 environment [23,24,25]. In this study, staff turnover continued to be a problem post-COVID-19. This resulted in the need for clinical resources to onboard new staff, as well as additional training and technical assistance from the study team. An additional major challenge this study identified was the paradox of valuing the technology to support patient care and the time it took for patients to engage with the technology, which at times had an adverse impact on this clinic’s workflows. While not identified in the current study, it is important to be aware of additional barriers to mHealth interventions that include digital literacy, staff training, and access to necessary equipment [25,26], as well as data security concerns [27].

Consistent with prior evaluations of the Health-E You in SBHC settings [18], this study showed the app’s potential to increase in non-barrier contraceptive use by nearly 86% (from 23.08% at baseline to 42.86% in interest after using the app). It is important to note that this increase assumes that the clinicians were able to provide the method of interest given that contraceptive care is part of their clinic’s delivery system. It is also important to note that most of the sexually active females indicated they were not ready to select a method of contraception even after using the app. This shows that this type of technology can support clinicians to better understand their patients’ needs and preferences through a more in-depth and individually tailored conversation about their needs, interests, and questions. As an implementation evaluation, this study did not assess post-knowledge changes; however, it did find variations in knowledge that digital tools can help address.

There are several noteworthy limitations of this study. It is important to emphasize that as a pilot implementation evaluation, it consists of a very small sample of clinicians and staff from one SBHC. While this was a small-scale pilot, it makes an important contribution to the growing body of literature by providing a focused analysis of implementation barriers and facilitators within a single SBHC. In addition, we were unable to gather data on youth who did not use the app due to limitations in the bandwidth of clinic staff to collect and provide this data. Further, we did not assess attitudes of the adolescent patients regarding their use/non-use of the technology; however, these attitudes have been reported elsewhere for the Health-E You app [15] and for RAAPS [28]. An additional limitation is the unique setting of this SBHC—it is located in California, with confidentiality protections for adolescent patients seeking SHR care without parental consent. As such, the generalizability of these findings is limited, and they should be interpreted cautiously.

The findings emphasize the necessity of addressing workflow interference and technical limitations, which may impede broader adoption. For example, integrating real-time alerts into their existing electronic health record and reducing the length of the assessment process could significantly improve the platform’s usability and acceptance. Additionally, these results support prior recommendations for comprehensive staff training, as highlighted in studies by Borges do Nascimento et al. (2023), which noted that adequate preparation is critical for ensuring the sustained success of digital health tools in youth-focused settings [29]. This pilot is critical as it allows programs to address these barriers before scaling up the intervention. 

Despite the barriers encountered, the platform’s perceived educational value and its ability to prompt discussions about sensitive SRH topics indicate promise for continued use. The staff’s unanimous support for renewing the license suggests that, with refinements, this approach could effectively enhance SRH care for AYAs. In considering the implementation of technologies in clinical practice, it is important to understand and address barriers. For instance, given the limited visit time clinicians have with their patients and the need to get them roomed and screened quickly, it is important to assess the accessibility and feasibility of alternative strategies for using technology beyond waiting to see the clinician from the perspectives of clinic staff and patients. SBHCs face the added challenge of seeing patients quickly during the school day so they can return to class as soon as possible. Ideas to consider include using the technology outside of the clinical setting, providing opportunities to use health technology as part of the health education curriculum, or allowing more time for the visit to allow for the use of technology; however, confidentiality must be protected for sensitive health topics, such as SRH, especially for AYAs. To address technology challenges faced by busy front staff and patients, clinics can consider the use of health-educators, patient navigators, or trained volunteers to provide support; however, this solution may require additional resources, which may be difficult, especially for safety-net clinical settings. A major challenge this study faced was the additional time it took for clinicians to receive and review the information the patient provided. While PDFs can be uploaded into the electronic health record (EHR), greater integration of this type of information in the EHR could enhance efficiencies by making the information more accessible and streamlining workflows. However, the cost:benefit ratios of such integration are important to consider, and more research is needed in this area [30]. Future research should also explore the scalability of this system across multiple SBHCs and examine strategies for overcoming implementation challenges.

In summary, this pilot study demonstrates that integrating RAAPS and the Health-E You app into SBHCs is feasible and valuable but requires targeted efforts to overcome operational and technical barriers. By refining these tools and their implementation processes, they hold potential to bridge SRH care gaps and improve health outcomes for AYAs. This study provides a critical framework for future mHealth interventions aimed at reducing health disparities through innovative, school-based healthcare delivery models.

## 5. Conclusions

This pilot study demonstrated that use of the RAAPS and Health-E You apps in a school-based health center setting is valued by clinic staff for enhancing sexual and reproductive health (SRH) education and care among adolescents. Despite operational barriers such as workflow disruptions, technical limitations, and staff resource constraints, the tools increased interest in non-barrier contraceptive methods and facilitated more tailored, in-depth SRH discussions with clinicians. These findings offer a valuable framework for future implementation efforts and underscore the potential of mHealth interventions to reduce health disparities in underserved youth populations.

## Data Availability

The data presented in this study are available on request from the corresponding author due to the privacy protections of participants.

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
