# Peer review of "A Pilot Study of Integrated Digital Tools at a School-Based Health Center Using the RE-AIM Framework"

_healthcare, 2025, doi:10.3390/healthcare13151839_

Round 1
Reviewer 1 Report
Comments and Suggestions for Authors
See comments attached.

Specific comments are included in the report attached. The paper needs editing.
Author Response
Reviewer 1 Comments |
Response |
Abstract1. The introductory sentence, “Adolescents and young adults (AYAs), especially those from underserved communities, often face barriers to sexual and reproductive health (SRH)”shouId be revised for greater clarity and specificity. The authors should clarify whether they are referring to barriers in accessing SRH services, or to the fact that this population experiences poorer SRH outcomes. Being explicit about the nature of the barriers will help strengthen the opening and better contextualize the focus of the study. |
We have clarified this 1st sentence as suggested and have specifically mentioned disparities faced by Black and Hispanic women. |
2. The methods section provides a helpful overview of the intervention and the data collection approach; however, several important elements need further clarification and elaboration to strengthen the rigor and transparency of the study:• Study Design Clarification: While the section identifies this as a pilot study, it does not clearly specify the implementation study design (e.g., mixed methods, hybrid designs, step wedge, quasi experimental etc). Naming the specific design would help readers understand the methodological framework.• Participant Information: There is no information on participant recruitment, sample size, inclusion and exclusion criteria, or consent procedures. Since both quantitative and qualitative data were collected, it's important to clarify how many participants contributed to each component and how they were selected.• Setting and Context: The setting (e.g., clinic type, geographic location, or demographic context) is not mentioned. Providing this information is essential for understanding the context of implementation and assessing generalizability.• Data Analysis: The section mentions that thematic analysis was conducted, but it lacks details about the process. Who conducted the analysis? Was coding conducted independently by multiple researchers? How was inter-raterreliability addressed (if at all)? Also, there is no mention of how quantitative data were analyzed, if at all. |
We have clarified in the methods section that this is a pilot implementation study consisted of pre-post app use data and a case-study where we examined, how one clinical practice site implemented the intervention using interviews and clinical observations. Because this is an implementation study, the participants are clinicians and clinic staff. De-identified data from youth participants was extracted to identify the number of users during the trial period. We have added a subsection, Setting/context and provided additional information about the clinical setting in the revised manuscript. Specifically, in addition to stating that this SBHC is in Los Angeles, we add that this clinic in an area of high need and serves primarily Latine/Hispanic youth. We have also specified the data analytic steps that we used. |
3. The sentence in lines 26-27 that “Between April 2024 and June 2024, a total of 14 students engaged with 26 the RAAPS screener and 10 completed the Health-E You app.” Should be moved to the methodology section. This is not a finding; it is just stating what was happened and when. |
Thank you for this recommendation, we have moved it accordingly. |
Introduction 1. In lines 49—51, the sentence stating, “Those from groups who have been excluded in care often come from underserved communities, and experience disproportionately higher rates of adverse sexual and reproductive health (SRH) outcomes,” lacks clarity. The authors should specify which groups they are referring to. Since reference [2] cited, identifies minority women and adolescent females of all races and ethnicities as being disproportionately affected, it would be more accurate and informative for the authors to explicitly mention these populations. Additionally, the sentence should be restructured for improved readability and precision. |
We appreciate the opportunity to provide additional clarity on the disparities in SRH outcomes and report this so that it is consistent with reference #2 as recommended by the reviewer. |
2. The authors should clarify the specific age group they are referring to, as definitions of age ranges can vary across studies and contexts.
|
We appreciate this comment and have added more information about the age group in the background and more details about the app participants in the results. |
3. The authors need to clarify the time period or specific study from which the figures in lines 55-56 are drawn. As written, the sentence feels incomplete, as it references a specific timeframe that is not explicitly stated. Readers should not be expected to search through the cited references to determine the relevant period, this information should be clearly presented within the text. |
The timeline of the study is now clearly defined in the methods section. |
4. In lines 57—58, the term “regular screening” is broad. It would be helpful for the authors to specify what types of screening are being referred to. Providing examples of relevant screenings, such as STI testing, cervical cancer screening, or contraceptive use, and more importantly the screening targeted in the pilot study being reported would add clarity and strengthen the reader's understanding of what is considered critical at this stage. |
Thank you for the opportunity to add more information on routine/regular health screenings. We have included this information in the revised manuscript. |
Methodology 1. The study design is not clearly stated. The authors should specify whether the study employed a hybrid, quasi-experimental, pre-post, mixed methods, or another design. Clearly identifying the study design is essential for understanding the methodological approach and interpreting the findings. |
As noted previously in the overall comment about the methods, we have clarified the study design. |
2. Ethical Considerations: The section does not mention whether the study received ethical approval or how informed consent was obtained. Even for a pilot study, these are critical components that need to be reported.
|
In addition to providing this information to the editorial team and at the end of the manuscript in additional details section; we have added that “The study received human use approval from the California, San Francisco, Internal Review Board” in the methods section
|
3. The authors need to clarify that two distinct participant groups—students and healthcare workers, were involved in the study, and clearly explain how each group was recruited. Additionally, the inclusion and exclusion criteria for each group should be specified. The sampling method should also be described, along with information on whether and how informed consent was obtained from participants.
|
Because this is an implementation study, the participants are clinicians and clinic staff. We have clarified that only de-identified data from youth participants was extracted for the sole purpose of identifying the number of app users during the trial period. Because this is an evidence-based practice tool that is being implemented in a real-world setting, apps users are not participating in a research study and their use of the app is part of regular clinic practice.
|
4. The authors should specify how many participants were involved in the quantitative and qualitative components of the study, respectively. Additionally, the rationale for determining these sample sized should be provided, including any consideration or calculations used to justify their adequacy, or any previous sources cited that used similar approach. |
We have clarified in the methods section that all clinic staff and clinicians were invited to participate. This was in order to capture the perspectives of everyone at this SBHC. To assess the reach of the app at this clinic site (as part of the Re-Aim framework), clinic staff were instructed to offer the app to all adolescent females (14-19 years of age). |
5. How was the quantitative data collected? Which implementation science measures were used? It is important to describe the measures and their sources. By the way, each component of the framework is a construct and had a measure associated with it that the authors could use to measure the implementation of their intervention. |
We have clarified that de-identified data was gathered from the back-end of the app which provided information on race/ethnicity, sexual activity status, method of current contraceptive use, app recommendations for contraceptive use, and the method the sexually active adolescents’ were most likely interested in using after completing the app (including the option, “I’m not ready to select a method.” This is provided in the results. |
6. How was quantitative data analyzed? Was any software used? |
As noted in the paper, descriptive analyses were used. Because of the small sample, no statistical software was used. |
7. How was qualitative data analyzed? There is need to describe the steps followed in this thematic analysis. Also, identify the analytic approach used, and give references. The description of the qualitative data analysis appears to be insufficient. More information is needed about the research assistant involved, specifically, their qualifications and role in the analysis process. It is unclear why the authors did not participate in the data analysis themselves or how they ensured the rigor and trustworthiness of the findings. The authors should clarify whether the data were validated and whether any experts reviewed or verified the interpretations to strengthen the credibility of the analysis. Was member checking of the data analysis done? |
The revised manuscript provides additional details on how the coding scheme was developed and how the qualitative data was analyzed. |
8. The authors should consider including a visual representation of the RE-AIM framework to illustrate how it was applied in the study. A figure or diagram would help readers better understand the framework's integration into the research design and analysis. |
Thank you for this suggestion. We have added in a figure of the Re-Aim framework and provided the reference for this figure. |
Results 9. There is need to describe the sample. Authors can add a sociodemographic characteristics table. |
The results section states that all adolescents were from Hispanic backgrounds and we have added in the mean age of the participants. |
10. Some information, such as lines 207 to 207 needs to be put in the methodology section. Any information related to how and what kind of data was collected, any information regarding how the intervention was implemented, how the participants engaged with the apps and how often, needs to be put under methodology. The results section should include findings related to reach, effectiveness, adoption, implementation, and maintenance. |
Thank you for this suggestion, as a result, we have reorganized the methods and the results. The results are significantly improved due to the suggestion reorganization related t0 the Re-Aim framework. |
11. The sentence in line 213 that “Clinic site visits and technical assistance calls corroborated these main themes” is not clear. Please clarify.
|
We have clarified this sentence to read: “Many of the themes that emerged during the interviews were also observed during the two site visits to the clinic and were documented and addressed during the T.a. calls.” |
12. Each quotation presented should be accompanied by a brief description of the individual who provided it. Since quotations reflect the perspectives of specific participants, it is important to include identifying details, such as role, age group, or other relevant characteristics in parentheses at the end of each quote to provide context and enhance the credibility of the findings. |
We have gone back through the interviews and provide the staff role for each of the quotes. |
13. The methodology indicates that both qualitative and quantitative data were collected; however, the findings from the quantitative component are not presented. Additionally, there is no integration of the quantitative and qualitative results. If the authors intend to report only the qualitative findings, this should be clearly stated. Otherwise, in a mixed methods study, both sets of data need to be presented and meaningfully integrated to provide a comprehensive understanding of the research questions. |
We have revised the manuscript to include more details on the quantitative analysis. See Table1 and expanded results section. |
Discussion 14. In some sections—such as lines 311—312—the authors repeat findings rather than engaging in a critical discussion. The discussion section should focus on interpreting and contextualizing the results, rather than restating them. |
We agree with the reviewer’s assessment and have provided a more critical discussion of the study findings. |
15. The limitation regarding the small sample size should be framed with caution, as this was a pilot study, and a smaller sample size is typically expected in such contexts.
|
Thank you for this recommendation, we have provided this in the limitation section of the discussion |
16. The recommendations section should go beyond simply listing suggestions; the authors should discuss how these recommendations are grounded in the study's findings. Moreover, some of the points currently labeled as recommendations appear to be next steps for future work rather than actionable recommendations based on the results. These should be placed in a separate section—either within the discussion or immediately following the recommendations.
|
We have made a distinction between implications of the research findings and future recommendations. |
17. The authors should also consider including a section on lessons learned from the study, as this can provide valuable insights for future research and implementation efforts.
|
We appreciate this suggestion and have expanded the discussion accordingly. |
18. The manuscript would benefit from a formal conclusion section rather than simply providing a summary. A conclusion allows the authors to succinctly restate the key findings, reflect on their significance, and emphasize the study's contributions to the field. Unlike a summary, which tends to reiterate what has already been said, a well-crafted conclusion provides closure and offers insights into the broader implication of the study, as well, as potential directions for future
|
|
References19. The references in the reference list should be formatted consistently. Currently, some article titles are written in title case (capitalizing most words), while others use sentence case (capitalizing only the first word and proper nouns). The authors should ensure uniformity in title formatting in accordance with the required citation style. |
Thank you for noticing the inconsistencies in reference formatting. We have reviewed the references and made formatting changes based on how other articles are published in this journal. We defer to the editorial staff to make any additional edits/recommendations to the format of the references. |
20. In the reference list, Reference No. 1 includes an issue number in parentheses, which may not be appropriate depending on the style being used. The authors should revise this reference to match the correct format based on the selected citation style guidelines.
|
We have revised the references accordingly. |

Reviewer 2 Report
Comments and Suggestions for Authors
“A Pilot Study of Integrated Digital Tools at a School-Based Health Center Using the RE-AIM Framework” assesses the implementation of two digital tools at a school-based health center, located in Los Angeles, to improve sexual and reproductive health (SSR) among adolescents, especially in underserved communities.
The introduction is a mixture of epidemiological background, care context and description of tools without a clear transition between ideas.
Introduction:
- Unplanned pregnancy statistics are cited without contextualizing the source or interpreting their relevance to the reader. Authors directly relate to limited access to services or the need for digital solutions.
- Description of the problem is redundant. For example: Ideas such as “limited sexual health knowledge, stigma, and access to care” are repeated in more than one section. Gaps should be condensed into a single paragraph and authors should delve into one or two of them.
- Presentation of the tools is premature and technical, which is the case of the Health-E You app. It is technically described (Likert scales, algorithm, WHO criteria) in the introduction. However, it is inappropriate for this section. This description must be included in the methods section when the intervention is presented. Thus, authors must refer briefly to its purpose and previous evidence.
- There is no a clear statement on the knowledge gap, as authors do not explain why it is necessary, since there are already other apps for the intended purpose.
Material and methods:
- The study design does not explicitly mention the approach used. It seems that it is a mixed methods research design, but this is not stated explicitly. I suggest that authors should make clear whether the methodological approach was mixed (exploratory or convergent) or simply complementary.
- It is mentioned that semi-structured interviews and clinical observations were conducted, but the exact number of observations and interviews is mentioned in the results, and not in this section. Thus, authors must specify the number of interviews and students in this section.
- It is not explained how authors determined that 14 students and 3 interviews were sufficient to assess the pilot study. Authors must include a brief explanation why they chose this number of participants. For example, they may point out that this number was considered due to theoretical saturation for interviews or environmental limitations.
- Authors must clearly specify that the interviews were conducted with staff, while students used the app.
- It is not clear whether there were inclusion and exclusion criteria, so these should be presented.
- The study involved minors, as participants were aged 14 to 19. In some countries, informed consent from parents or legal guardians is required for minors. Therefore, it is necessary to specify whether this requirement is required in the country where the study was conducted.
- In the analysis of the semi-structured interviews is mentioned that "thematic analysis" was done, but the process is not explained: was any software used? How many coders participated? Was there cross-validation?
- Authors do not detail how the quantitative and qualitative data were integrated. Thus, it is necessary to explain whether they were used in parallel, sequentially, or integrated in the analysis.
- Authors neither mention what happened when a student did not complete both parts of the tool (RAAPS + App) nor how this was considered in the analysis. Therefore, authors must make clear in the analysis or data collection the criteria used to define "complete use" and how non-response was addressed.
- Authors must specify whether the semi-structured interview guide was designed for the study or derived from previous studies. Also, they must state whether the guide was previously approved.
Results
- Results are reported considering only 14 students, but it is not explicitly stated that these findings are not generalized. Although this is a pilot study, it would be helpful to include a statement in this section indicating the sample size limitation.
- Data presented are basic (absolute numbers), without additional descriptive statistics such as percentages or measures of central tendency. For example, “10 students completed the app”; or: “which represents 71% of the total.” Authors must include basic proportions or visualizations to facilitate interpretation (tables, bar charts).
- It is mentioned that the RE-AIM framework was used, but the results are not explicitly organized by its components (Reach, Effectiveness, Adoption, etc.). thus, this section must be reorganized or subtitle it to reflect the five elements of the framework and improve methodological consistency.
- Poor systematization of the qualitative analysis. Topics are identified, but it is not mentioned whether they were coded and apparition frequencies are not shown. Thus, authors must include tables containing emerging topics, codes, and exemplary quotes, or at least a thematic frequency summary.
- It is stated that three interviews were held (nurse, medical assistant, reception staff), but it is not clearly specified how they were selected or how representative their testimonies are. Thus, authors must include in the results how many interviews and observations were conducted and whether theoretical saturation was reached or illustrative testimonies were obtained.
- It is mentioned that clinical observations were conducted, but the results do not describe what was observed, how, or what patterns were detected. Authors must include at least a summary of the observational findings to enrich data triangulation.
- It is not analyzed whether there were differences in the experience or use of the tools according to age, schooling level, previous sexual experience, etc. Although the sample is small, authors must indicate whether they observed any trends or if they were not able to analyze them due to the sample size.
- Only two ratings of the app are reported (5 and 3 stars), without further qualitative analysis of the adolescent user experience. Thus, authors must elaborate or state that further research focusing on end-user voice is needed.
- The separation between results and discussion is not approppriate, as some sections interpret the results or express opinions.
Discussion:
- The discussion begins by reiterating findings without a deeper interpretation or critical synthesis. Thus, authors must start this section with a key statement or general insight contextualizing the value of the study value and how it is related to the existing literature.
- The discussion mixes operational, technical, and contextual topics without a clear separation or subtitles, which makes difficult to follow the core of the argument.
- Although some previous studies are cited (e.g., RAAPS, Health-E You), the comparative analysis is superficial. Thus, authors must delve into how the results agree or differ with previous studies, as well as interpret possible reasons (context, population, school environment, etc.).
- Technical and operational gaps are noted, but discussion of education value and user acceptance is minimized. Therefore, authors must keep the discussion balanced by focusing on facilitators, impact potential and how to exploit them more effectively.
- The effects of COVID-19 are mentioned, but the way they directly influenced this study is not explained.
- Ethical aspects to implement digital sexual health tools among adolescents is neither discussed nor possible cultural or gender gaps.
- The limitations are briefly addressed nearly the end and without breaking down their implications.
- Although some recommendations are delivered, they are not prioritized or directly linked to the findings.
- The conclusion of the section is lengthy, weak and the message is not clear.
References:
- For some references the titles are abbreviated or incomplete, so they are difficul to locate.
- In reference 13 (National Academies Press), the collective author is badly presented: “Committee on Systems Approaches to Improve Patient Care by Supporting Clinician Well-Being.” Instead, it should be presented as an institutional author National Academies of Sciences, Engineering, and Medicine.
- Old studies (for example, one from 1997) are used to justify digital tools, when there is more recent literature on mHealth and SRH.
- Inadequate citation of software and platforms, as no formal reference for RAAPS or Health-E You as a digital tool is provided.
- Although most articles include a DOI, in some of them it is missing or is not uniformly presented (e.g., “https://doi.org/”).
Overall, in my opinion this article must not be accepted for publication in its current version. Although it addresses a relevant topic—implementing digital tools to improve adolescents' sexual and reproductive health in school settings—it is a pilot study with serious methodological limitations that compromise the validity of the findings.
Some of the main weaknesses identified: the sample is very small and there is no explanation, unclear inclusion criteria, poor systematization of qualitative analysis, ambiguity when integrating mixed data and a general structure that makes the study difficult to understand. Furthermore, the discussion lacks critical depth, while the references are inconsistent in form and substance.
Given that these limitations are substantial and affect both the design and interpretation of results, I therefore recommend that this manuscript be rejected. However, I suggest that the authors consider restructuring the study, strengthening its methodology and analysis and resubmitting it once they have corrected the noted deficiencies.
Round 2
Reviewer 2 Report
Comments and Suggestions for Authors
"No comments"
Comments on the Quality of English LanguageAlthough the English is generally understandable, there are several passages with dense writing, minor punctuation errors, repetitions, and long or ambiguous constructions that hinder fluent reading.
Author Response
I reviewed and accepted the recommended changes that the MDPI editor made, fixed a couple of typos, and indicated that Steven Vu is the corresponding author.